# Highly Skin-Conformal Laser-Induced Graphene-Based Human Motion Monitoring Sensor

**DOI:** 10.3390/nano11040951

**Published:** 2021-04-08

**Authors:** Sung-Yeob Jeong, Jun-Uk Lee, Sung-Moo Hong, Chan-Woo Lee, Sung-Hwan Hwang, Su-Chan Cho, Bo-Sung Shin

**Affiliations:** 1Department of Mechanical Engineering, The University of Tokyo, Tokyo 113-8656, Japan; ysjsykj8025@naver.com; 2Department of Cogno-Mechatronics Engineering, Pusan National University, Pusan 46241, Korea; lju3534@naver.com (J.-U.L.); cwleeho2@naver.com (C.-W.L.); po78765@naver.com (S.-H.H.); cho_brian@naver.com (S.-C.C.); 3Interdisciplinary Department for Advanced Innovative Manufacturing Engineering, Pusan National University, Pusan 46241, Korea; hsm14789@naver.com; 4Department of Optics and Mechatronics Engineering, Pusan National University, Pusan 46241, Korea

**Keywords:** laser-induced graphene, wearable sensor, motion monitoring, photosensitive polyimide, polydimethylsiloxane

## Abstract

Bio-compatible strain sensors based on elastomeric conductive polymer composites play pivotal roles in human monitoring devices. However, fabricating highly sensitive and skin-like (flexible and stretchable) strain sensors with broad working range is still an enormous challenge. Herein, we report on a novel fabrication technology for building elastomeric conductive skin-like composite by mixing polymer solutions. Our e-skin substrates were fabricated according to the weight of polydimethylsiloxane (PDMS) and photosensitive polyimide (PSPI) solutions, which could control substrate color. An e-skin and 3-D flexible strain sensor was developed with the formation of laser induced graphene (LIG) on the skin-like substrates. For a one-step process, Laser direct writing (LDW) was employed to construct superior durable LIG/PDMS/PSPI composites with a closed-pore porous structure. Graphene sheets of LIG coated on the closed-porous structure constitute a deformable conductive path. The LIG integrated with the closed-porous structure intensifies the deformation of the conductive network when tensile strain is applied, which enhances the sensitivity. Our sensor can efficiently monitor not only energetic human motions but also subtle oscillation and physiological signals for intelligent sound sensing. The skin-like strain sensor showed a perfect combination of ultrawide sensing range (120% strain), large sensitivity (gauge factor of ~380), short response time (90 ms) and recovery time (140 ms), as well as superior stability. Our sensor has great potential for innovative applications in wearable health-monitoring devices, robot tactile systems, and human–machine interface systems.

## 1. Introduction

Over the past decades, strain sensors have been developed for industrial safety, architectural accuracy, and micro processing [1,2,3,4,5]. Recently, the importance of strain sensors has been increasing with the leap into the hyper-connected age of the 4th industrial revolution. It is possible to receive data from the human body; these data are the most important part of connectivity between human and technology and involve using strain sensors [6,7,8,9,10,11,12,13]. Therefore, it is necessary to develop a strain sensor that has good contact with the body and is thin, hydrophobic, flexible, and similar in color to human skin. Such a sensor will be directly attached to the human body, handicapped prosthetic limb, or robot. Various attempts have been made to use polydimethylsiloxane (PDMS) as a substrate for a strain sensor [13,14,15,16,17]. As PDMS has good advantages such as hydrophobicity, moldability, low interfacial free energy, and thermal and chemical stability, it is actively used as biocompatible substrate of a strain sensor [17,18,19,20,21,22]. Carbon-based nanomaterials are actively applied as electrodes of these strain sensors. Among such nanomaterials, graphene has received a lot of attention. Graphene is a single layer of carbon and has a unique structure, like a honeycomb made of sp^2^ bonds [23,24,25]. Graphene has a variety of advantages such as strong physical properties with a tensile strength of 130 GPa and an elastic modulus of 1 TPa, as well as high thermal conductivity and optical transparency [25]. Graphene can be fabricated via a variety of fabrication methods such as dry exfoliation [26,27,28], epitaxial growth [28,29,30,31,32], and chemical vapor deposition (CVD) [32,33,34]. In terms of mass production, various processes have been developed, but it is tricky to apply graphene’s remarkable properties to sensors because of the Van der Waals forces based on the 2-D structure [35]. These cause aggregation [36], restacking [37], and disorder [38] in the graphene sheets and result in dramatically reduced surface area. Recently, 3-D porous structures of graphene (3-D PGS), called “Graphene aerosol”, have been reported to have better properties than those of 2-D graphene [39,40,41,42]. Therefore, many researchers have attempted to fabricate 3-D PGS. Tour et al. published several papers on laser induced graphene (LIG) based on 3-D PGS formed by irradiating a polymer using a CO_2_ infrared laser with a wavelength of 1064 nm [43,44,45,46,47]. These types of LIG exhibit properties similar to those of graphene and are applicable to various fields. Tao et al. reported LIG fabricated using a 450 nm visible laser for application to sound-sensing [48]. Carvalho et al. presented a sensitive strain sensor utilizing LIG produced by UV irradiation [49]. Shin’s group also investigated the characteristics of LIG fabricated using a 355-nm UV pulsed laser and applied to physical and electrochemical sensors [50,51,52]. LIG has been proven through recent research that a significant number of both pentagonal and heptangular carbon rings are formed in the structure by laser irradiation on the polymer, forming a three-dimensional carbon structure in which carbon rings are entangled [53]. Since graphene aerosol is not only flexible but also highly sensitive to various changes in electrical properties stemming from physical changes, this material tends to be applied to human motion, voice, breathing, and pulse sensing. In this paper, we present a new fabrication method for a superbly sensitive skin color-like human monitoring strain sensor based on LIG fabricated by UV pulsed laser. We conducted an experiment to manufacture LIG by mixing liquid photosensitive polyimide (PSPI) into PDMS and irradiating the mixture with a 355 nm UV pulsed laser. Unlike conventional polyimide-based LIG sensors or other nanometal wire sensors, this skin color-like LIG sensor has advantages such as flexibility, obtained by using PDMS substrate, and high sensitivity to physical elongation, obtained by using an LIG electrode, which induces a piezoresistive effect. The most important part of this skin color-like LIG patterned sensor is that it can be easily attached to many parts of the human body (biceps, elbow, finger, and throat). The attached sensor was able clearly to detect movements (strain: 40%) according to bending and stretching of the body; electrical properties of the sensor according to usage of sounds were also different, so that it can be used for voice detection. As a result, because our fabrication method is very simple, this sensor has high potential for medical and athletic wearable electronics and promising applications in a variety of fields.

## 2. Materials and Methods

### 2.1. Preparation of Skin Color-Like Polymers (SLPs)

All prepolymers were prepared in liquid state for uniform dispersion. PSPI (PIP 100, PNS technology, Anyang, Korea) was added to PDMS base and its cross-linking agent (Sylgard 184, Sigma-Aldrich, St. Louis, MO, USA) and then magnetically stirred for about 5 min. The mixtures were prepared by adjusting the weight ratio (PDMS:PSPI = 20:1, 10:1, 5:1, and 5:2). As-prepared mixtures was transferred onto silicon wafer (4SWP06, Prime grade, Silicon Technology Corp, Tokyo, Japan) and then spin-coated at 500 rpm for 1 min. Subsequently, the sample was cured for 1 h at 200 °C on a hot plate and then cooled to room temperature.

### 2.2. Electronic Skin Fabrication via UV Pulsed Laser System

Facile Laser Direct Writing (LDW) was adopted for the preparation of active materials of the skin color-like strain sensor. A 3-D PGS of LIG was fabricated using a 355 nm pulsed laser (AONano 355-5-30-V from Advanced Optowave, Ronkonkoma, NA, USA) under ambient condition. Laser beam trapping was performed through the movements of the mirrors of the galvano scanner (HurrySCAN III 14 from SCANLAB, Pucheim, Germany) and the F-θ lens, f: 105.9 mm (S4LFT4100/075 Telecen-tric Scan Lens from Sill Optics, Wendelstein, Germany).

### 2.3. Characterization

The morphology of the LIG of the sensor was analyzed by field emission scanning electron microscope (FE-SEM, TESCAN MIRA 3 LMH In-Beam Detector, Brno, Czechia) imaging. For the elemental analysis, an energy dispersive X-ray spectrometer (EDS, TESCAN MIRA 3 LMH In-Beam Detector, Brno, Czechia) was also utilized. Raman spectra were obtained using a Micro Raman spectrometer (JASCO NRS-5100, Easton, MD, USA) with a 532 nm excitation laser system. Fourier transform infrared spectroscopy (FTIR) spectra were recorded in the range 500–3500 cm^−1^ employing a JASCO FT-4100 (Easton, MD, USA). For the analysis of the composition and chemical bond state of the LIG patterns, X-ray photoelectron spectrometer (XPS) spectra were analyzed with a THERMO VG SCIENTIFIC Multilab 2000 (Waltham, MA, USA). To obtain the crystalline characteristics, XRD was conducted on an X’Pert-MPD System (PHILIPS, Amsterdam, The Netherlands) with Cu Ka radiation (λ = 1.54 Å). Instantaneous resistance measurements were performed using an LCR meter (LCR Meter, Keithley Tektronix, Beaverton, OR, USA) at a DC voltage of 1 V.

### 2.4. Formula

For the initial length L_0_ and secondary length L_s_, the strain is defined as follows:(1)$$\;\varepsilon \;\left(\%\right)=x=\left|\frac{L_{s}-L_{0}}{L_{0}}\right|\times 100\%$$

A common representation of the sensitivity of the sensor’s electrical response to mechanical changes is the gauge factor, which depends on the electrical resistance of the sensor when strain is applied:(2)$$\mathrm{GF}=\;\left|\frac{R\left(\varepsilon \right)-R_{0}}{\varepsilon }\right|$$
where R_0_ and R(ε) are the initial electrical resistance and the electrical resistance in the presence of strain, respectively.

The average value of the hysteresis (h) was calculated from the resistance (R) values, as follows:(3)$$h\;\left(\%\right)=x=\;\left|\frac{R_{S}-R_{r}}{R_{m}}\right|\times 100\;\%$$
where the subscripts s, r, and m represent the stretching, release, and maximum value of the resistance at a particular strain, respectively.

The Simmons equation is defined as:(4)$$R_{\mathrm{tunnel}}=\frac{V}{\mathrm{AJ}}=\frac{h^{2}d}{{\mathrm{Ae}}^{2}\sqrt{2m\lambda }}\cdot \exp\left(\frac{4\pi d}{h}\sqrt{2m\lambda }\right)$$
where J is the tunneling current density; V is the electrical potential difference; A is the cross-sectional area of the tunnel; e is the quantum value of electricity; m is the mass of an electron; h is Planck’s constant; d is the distance between conductive particles; and λ is the height of the energy barrier.

The crystalline size along the a axis is calculated from the ratio of the intensity of the G peak to that of the D peak in the Raman spectrum. L_a_ can be obtained from the following equation:(5)La=2.4 ×10−10×λR×(IGID)
where λ_R_ is the wavelength of the Raman laser (λ_R_ = 532 nm).

The domain size (L_a_) and crystalline size (L_c_) of LIG are calculated from the XRD characteristics using the following equations:(6)$$L_{a}=\frac{1.84\lambda }{B\left(2\theta \right)\cos\theta }$$
(7)$$L_{c}=\frac{0.89\lambda }{B\left(2\theta \right)\cos\theta }$$
where B(2θ) (in radian units) is the full width at half maximum of peaks (002) and (100) and λ is the wavelength of the X-rays (λ = 1.54 Å).

## 3. Results

### 3.1. Fabrication of Electronic Skin Based on LIG

Our flexible and stretchable electronic skin (e-skin) consists of 3-D PGS networks produced by direct laser scribing. The laser used in this experiment is a UV pulsed laser with a wavelength of 355 nm, which causes a photochemical ablation, resulting in a high surface area porosity on the surface of the laser-induced graphene specimen. The fabrication process is shown in Figure 1a. Skin color-like polymers (for details, refer to Experimental Section 2.1) were spin-coated to give the samples constant thickness of from tens to hundreds of micrometers, with uniform dispersion. Depending on the different curing temperatures of PDMS (100 °C) and PSPI (250 °C), we baked the PDMS first and then applied a hot plate to maintain the temperature at which the PSPI could cure completely. When baking at the curing temperature of PSPI from the beginning, because the heat resistance of PDMS is exceeded, noticeable pores appear after gases radically evaporate from the sample, and the flexible PDMS itself loses its properties. What is noteworthy here is that the skin color can be determined according to the composition of the PDMS/PSPI mixture. The color of the skin color-like polymer becomes darker as the PSPI composition increases. The polymers were placed in a defocused 355 nm laser beam under ambient condition. Skin color-like substrate was irradiated with laser scanning speed from 40 mm/s to 60 mm/s and laser power of 1.2 W to obtain LIG optimized for human-motion monitoring sensors. In the viewpoint of mass production, the time to fabricate a sensor platform by directly irradiating the prepared polymer with a laser is ~15 s. A scanning gap larger than the spot diameter was implemented to leave an unburned area between the two scan lines. On the other hand, a smaller scanning gap was used to overlap the carbonized areas, causing over-burning of the skin color-like polymers. Likewise, the high speed of the laser tip sparsely burns the polymer, causing less graphitization. Considering these experimental conditions, we adopted a scribing power of 1.2 W, a line gap of 0.6 μm, and a scanning speed of 60 mm/s. The e-skin was fabricated to produce a sensing area of 1 × 0.5 cm^2^. Electrical contacts were formed on both sides of the sensing area by connecting Cu wires with the help of silver paste. As can be seen in Figure 1b, the e-skin is skin-conformal and skin color-like. Appendix A shows the color of prepared samples according to the weight ratio of PDMS and PSPI. The surface morphology of e-skin fabricated according to the weight ratio of PSPI/PDMS was analyzed by FE-SEM in EDS mode, with results shown in Figure 1c and Appendix A. The FE-SEM image in the figure and its inset clearly show that UV laser scribing of skin color-like substrates results in a 3-D PGS with a nanostructure composed of a wrinkled flake-like morphology, resulting from the rapid liberation of carbonaceous and gaseous products. Appendix A shows laser-induced graphitic patterns with porous structure due to instantaneous emission of gas due to the generation of very high localized temperature during laser irradiation. The pore size of the structure is tunable by controlling the laser power, resulting in a rate and an amount of liberation of the gas [43,44,45,46,47]. Elemental mapping of the e-skin was in FESEM-EDS mode. Obviously, as the weight ratio of PSPI in our skin color-like polymer increases, SEM images show that, after laser irradiation, many micron-sized cotton-like constituents, mostly composed of carbon, silicon, and oxygen (Appendix A), are generated. This causes the volume occupied by the pattern per unit mass to increase, and thus the permittivity increases as the presence of wrinkled LIG flakes increases, giving the sensor better electrical properties. According to the EDS analysis results shown in Appendix A, the lower the weight ratio of PSPI, the lower the oxygen composition ratio in the graphitic pattern after laser irradiation. This is because the lower the PSPI ratio was, the higher laser fluence that was needed to irradiate the substrate, and the higher the localized heat generated at this time to quickly discharge the gas. Studies reveal that, because of oxidation in the laser-scribed zone, the fabrication of large-scale porous graphene is possible without need for an inert atmosphere. The flexible and stretchable e-skin can be fabricated through a simple one-step method of direct UV laser scribing. Many studies have successfully converted polyimide films and particles to LIG using various lasers, and so graphene formation is most likely a photothermal process in which high localized temperatures of >2500 °C can easily break the C-O, C=O, and C-N bonds of PI. This paper reports that PSPI is successfully converted to LIG by 355 nm pulsed laser irradiation. To further analyze the crystallinity, elemental composition, chemical interactions, and compositions of the active areas of e-skin based on LIG, the material was systemically studied by Raman spectroscopy, XPS, FTIR, EDS, and XRD, respectively. The typical Raman spectrum of carbon materials contains bands marked as D, G, and 2D [54]. The D peak at 1350 cm^-1^ is an indication of a disorder band, showing that the graphene layers had considerable defects, such as vacancies and strained hexagonal/non-hexagonal (pentagon or heptagon) distortions that result in corrugation and twisting of layers [55]. The two Raman signatures of graphene are the G peak at ~1580 cm^−1^ and the 2D peak at ~2700 cm^−1^ [54]. The G peak is related to in-plane vibration of the sp^2^ hybridized carbon atoms of graphene. The 2D peak is related to the number of and the stacking order of the graphene layers. Based on data obtained from the Raman spectra, additional information about the carbon materials can be obtained by analyzing the ratio of the intensities of individual peaks [56]. Figure 2a shows that the PDMS + PSPI model was graphitized after laser irradiation and clearly shows D, G, and 2D peaks. To analyze the crystal characteristics of LIG according to the ratio of PSPI/PDMS, we analyzed the intensity ratio and full width at half maximum (FWHM) of the G peak in Figure 2b or Appendix A. As a result of analyzing the Raman spectrum for each polymer model, when models 1 and 2 were graphitized after laser irradiation, the Raman spectra were similar to that of glassy carbon, as shown in Appendix A. However, the spectra from models 3 and 4 are obviously different from those of glassy carbon. The I_D_/I_G_ ratio is related to the number of defects present in the carbon-based material. Figure 2b shows that the 5:2 (PDMS:PSPI) weight ratio sample has the structure with the fewest defects. The ratio of the peak intensities I_2D_/I_G_ indicates the number of graphene layers in the material [57,58]. When the 2D band increases and the G band decreases, the number of layers in the material is assumed to increase. Therefore, the 5:1(PDMS:PSPI) ratio sample has the fewest graphene layers among all samples. We obtained the FWHM of the G peak and used it as a parameter for the estimation of the degree of development of carbon crystallities [59,60,61,62]. The G peak becomes narrower with the development of carbon crystallites in carbonaceous materials [61].

The sample with the 5:1 weight ratio is assumed to have the best crystalline characteristic, and this indicates that the 5:1 sample has better crystalline graphene formation and fewer layers of graphene formation. Additionally, the domain size (L_a_) of graphene aerosol was analyzed through I_D_/I_G_ analysis of each skin-polymer sample” OR “individual skin-polymer samples. Raman spectroscopy plays a pivotal role in obtaining the crystalline size along the a axis of a carbon material. Figure 2c shows the increase in the crystalline size (L_a_) to ~49 nm as the weight ratio of PSPI increases. These values are calculated from Equation (5) in Section 2.4, Formulas. Fourier transform infra-red (FT-IR) analysis was performed to investigate the molecular groups within the active surface area of the e-skin during the laser processing procedure. The patterns are similar to those of PDMS because of the high weight ratio of PDMS [63]. Among the patterns, the most intense ones obtained were those associated with asymmetric −CH3 stretching in Si−CH3 (2950–2970 cm^−1^), stretching vibration of Si−O−Si bonds (1000–1100 cm^−1^), −CH3 rocking, and Si−C stretching vibration in Si−CH3 (785–815 cm^−1^) [64,65,66]. By comparing the spectra of four of the as-prepared samples, as shown in Appendix A, it can be concluded that the absorption peaks at 704–760 cm^−1^ due to Si-C stretching become intense with increases in the weight ratio of PSPI, indicating that this is a good carbon-based material precursor for detecting strain [67,68]. It is obvious that all e-skin samples show an absorption peak at about 1073 cm^−1^, which is assigned to the C–O–C stretching vibrations of graphene [69]. The patterns show a small peak at 1264 cm^−1^ assigned to the C–C bonds [70]. The C-H stretching vibration peak of the methyl group associated with PDMS and graphene-based materials is observed at 2929 cm^−1^ [71]. However, this signal overlaps with the absorption resulting from the sulfonic acid group, which is in the same region [72]. It can be seen that LIG is partially formed on the skin color-like substrate. The chemical composition of the active materials contained in our sensing material was further confirmed by XPS spectra. The survey spectra acquired on the e-skin confirm the presence of C, O, and Si; all of these were also confirmed by the EDS spectra. As can be seen in Figure 2e, the XPS spectrum of our graphitic pattern exhibits two distinct peaks, an O1s peak and a C1s peak. Comparison of the five curves shows that the C1s peak rises as the PSPI ratio of the model increases. This is further confirmed by the high resolution C1s XPS spectrum of our laser-induced graphitic pattern, shown in Appendix A. This indicates that, due to UV laser irradiation, the C-C peak increased and the C=O and C-Si peak decreased, resulting in better quality graphene layers; these results agree well with the Raman spectra and XRD results.

We carried out XRD analysis of the skin color-like substrate and e-skin with variation of the PDMS/PSPI weight ratio, with results shown in Figure 2d. For the pure, skin-like substrate, there is a broad diffraction peak ranging from 18 to 80°. As we had speculated from the composition of the skin-like polymer substrate, we observed a characteristic broad peak at a 2θ of about 12 in the XRD pattern, indicating the amorphous nature of PDMS [73]. These results are compared with the e-skin’s active area, fabricated by laser irradiation of the substrate. The XRD patterns of the e-skin show similar characteristics. Broad diffraction peaks are observed at around 25°; these are related to the presence of a short-range order graphitic structure [74,75]. These three XRD patterns indicate that LIG flakes were uniformly localized on the surfaces of the skin-like substrate. The crystalline size along the c axis (L_c_) and a axis (L_a_, domain size) are calculated using Equations (6) and (7) from Section 2.4 (Formulas). These results are deduced from the characteristics of the XRD peaks and calculated as L_c_ = ~32 nm and L_a_ = 48 nm, respectively.

### 3.2. Working Mechanism of Skin-Like Electronic Skin

Strain sensors are electronic devices that can transform mechanical deformation into electrical signals; these sensors have been widely applied in wearable electronics and health care devices such as electronic skin (e-skin). There are a variety of kinds of strain sensor based on fiber Bragg grating, relative change of capacitance, and variation of light absorption; resistive strain sensors have been investigated intensively due to their ease of fabrication, compatibility with electronic circuits and excellent mechanical tunability [76,77,78,79,80,81]. 

As mentioned in the introduction, there are various methods of fabrication of resistive strain sensors based on graphene, which have excellent properties [49,50]. Strain sensors have been developed to have excellent stretchability and sensitivity by embedding graphene-based material within an elastic substrate such as PDMS and ECOFLEX. Although such sensors offer high Gauge Factors (GFs) and stretchability, durability and overshoot in the resistance response limit their practical applications in subtle human skin deformation. In this work, the naturally viscous material PSPI is used to give the substrate an excellent repeatable stretching property and good uniformity of the small thickness of the substrate for detecting small deformation from human motion. Our e-skin has similar mechanisms through which, when some external mechanical stimulation is applied, the internal conductive network becomes disconnected, and this results in resistance change. However, there is no limit of durability stemming from the mis-matched strain between the substrate and the active materials. This is because the sensing area was fabricated by irradiating a UV laser on the substrate itself. To further investigate the sensing mechanism of e-skin, the morphology variations of the sensing unit under strain were investigated. The strain (30, 60 and 90%) applied to the samples directly changes the morphological characteristics, as shown in Appendix A. It can be seen that graphene flakes moved away under applied strain, which could explain the change of the structure characteristics. Based on the unique properties of graphene nanosheets, the resistance of e-skin under small strain range is attributed to the change of the tunneling effect between the graphene sheets [82]. According to the Simmons function, the resistance of the e-skin can be calculated through the Equation (4) from Section 2.4 [83]. Under a small range of strains, the current pathway will be reduced within the e-skin intra-layer inside the graphene nano sheet and inter-layer in the overlapping areas of the graphene nano sheets. Some of the interconnections at the edges of graphene may also be broken [84]. By increasing the length of the e-skin, high variation of the electrical resistance of the multilayer graphene nano sheets results due to displacement of the layers and changing of the overlapping area and interruption of the layer connections. This mechanism may make it possible to explain that the change in GF according to the deformation of our e-skin is very linear. However, under a strain of approximately 120%, the large mathematical value of GF of ~380 calculated from the extremely high resistance change is ultimately the result of the complete loss of electrical connection (as can be seen in point F2 in Figure 3d). Therefore, the GF of the sensor that can be stably tested can be selected from the range of strain 0 to 110% and the GF in this case is ~110 (as can be seen in point F1 in Figure 3d).

### 3.3. Human-Motion Monitoring Using Electronic Skin

The piezoresistive properties of the graphitic patterns embedded in the skin color-like substrates were investigated for human motion monitoring by using a motorized tensile machine that was able to apply uniaxial tensile strain to the substrates. Appendix A shows the measurement system used to investigate the piezoresistive properties of the e-skin under tensile strain. Using an LCR meter and a source meter, the piezoresistive properties were recorded by applying a continuous voltage (1V) to the e-skin to measure the change of electrical resistance during tensile deformation of the sensor. The current-voltage (I-V) characteristic of e-skins for various strains (30, 60, and 90% strain) is depicted in Figure 3. Figure 3a–c shows the testing repeatability (over 10,000 times) at 30, 60, and 90% strain with frequency of 1 Hz. Since we fabricated e-skin by irradiating a UV laser on the substrate itself, there is no contact resistance between the sensing part and the skin color-like substrate, which would adversely affect the durability. Therefore, strong adhesion between the graphene flakes and the substrate causes graphene embedded in the substrate to experience the same strain level as the substrate. Our e-skin showed low electrical resistance hysteresis of 4% (at 90% strain). As can be seen in Figure 3e, the e-skin exhibited a response time and recovery time of ~90 ms and ~140 ms, very fast speeds compared to those of other wearable sensors [4,5,6,7,8,9,10,11,12,13,14,15,16,17,18,19,20,21,22]. When e-skin is subjected to strain, the gap between graphene-aerosol flakes increases linearly with increasing strain because of deformation of the PDMS/PSPI substrate. In the 90% strain repeatability test, it is presumed that the gap between the graphene-aerosol flakes, due to the Van der Waals force, widens because of electrical hysteresis. Therefore, after 10,000 iterations of the tensile strain test, the structure of the e-skin does not return to its initial state. In spite of this phenomenon, our sensor has high consistency of all strain-cycle responses under cyclic processes. In addition, as shown in Appendix A, the e-skin shows a low hysteresis effect at 30 and 60% strain, demonstrating its reliable durability and showing high possibility for use in wearable devices. All e-skin samples have relative resistance variation curves that ascend with increase in strain and fracture at ~125% strain, as shown in Figure 3d. Therefore, the strain sensing range of the e-skin is 120%, and this is entirely due to the physical properties of the skin color-like substrates. The value of GF is 21–35 in the strain range from 0 to ~1%, which indicates that our e-skin has high potential for application to wearable human motion detecting devices. Additionally, the value of GF reaches ~380 in the strain range of 115 to 120%, which easily exceeds those of other graphene family-based strain sensors [12,15,17,18,22,85]. Figure 3f shows that the body of the sensing area of the e-skin changes according to the application of strain, which affects the relative resistance. Additionally, by applying stress (which results in strain) in steps, with each application of force the V-I curve starts to shift.

### 3.4. Application of E-Skin for Human Motion Detection

To evaluate the potential of the skin color-like e-skin as a wearable device for human-motion and sound (from vocal cords) detection, the e-skin was attached to different skin parts of the human body (eyebrow, mouth, finger, biceps, wrist, and throat), as shown in Figure 4a. Since our e-skin shows large relative resistance changes to fine strain, it has excellent responsiveness in detecting changes in the human pulse, vocal cords, and facial expressions. Figure 4b,c exhibit responses of the e-skin to the raise–relax cycle of the eyebrows and to movements of the muscles near the lips. The sensing unit of the e-skin shows relative resistance changes with movement, which indicates its ability to sensitively detect subtle facial movements. This result could suggest the potential to apply this material as an emotion recognition sensor. Our e-skin showed great responses to joint and muscle movements. Figure 4d shows the relative resistance curve resulting from analysis of movement of bending a finger and indicates our sensor’s potential application in the field of finger-gesture control in robots and humans. Since the e-skin has response and recovery times of ~90 ms and ~140 ms, it is able to detect the strain created by even very fast finger movements. In another case, our e-skin was found to be capable of capturing subtle signals with high sensitivity under small strain changes caused by stepped-bending of a finger. Figure 4e shows that the skin color-like sensor has great sensitivity to bicep contraction and relaxation (two types: squeeze–relax and rotate–relax) over very short time intervals. The change in resistance due to muscle contraction caused by squeezing the muscle is more pronounced than the change in muscle contraction due to rotating, which shows the possibility that this sensor could make a contribution to a wearable system for muscle training or rehabilitation exercises. The response time and recovery time of the sensor are ~90 and ~140 ms, which are impressively fast compared to those of other strain sensors [5,6,7,8,9,10,11,12,13,14,15,16,17,18,19,20,21,22,54,55].

PDMS, the substrate of the e-skin, is not at all harmful to the skin and is flexible, and so the attached sensor is very suitable for skin and has high sensitivity to subtle muscle movements including the pulse of the wrist, as shown in Figure 4f. In modern times, measuring arterial pulse rate per minute, or heart rate, is very important. Therefore, most strain sensors measure the pressure wave generated by the heart through the wrist or neck, which is a general standard for strain sensors [57,58,59,60,61,62]. Analysis of pulse waves can be used to measure various health conditions of the human cardiovascular system. Figure 4f shows the response of the e-skin when continuously monitoring the pulse of a 28-year-old male subject. The pulse pressure wave induces deformation of the graphitic area, which is the sensing area of the e-skin; there is a gap between the 3-dimensional graphene flakes that leads to a distinct change in the relative resistance of the sensor. From the arterial pulse, a distinct change in electrical property of the e-skin was observed and it can be seen that the e-skin recorded 6 to 7 pulses over 5 s, which is 70 to 80 pulses/min. In addition, Appendix A shows that the e-skin can detect characteristics of the wrist pulse consisting of three peaks (P1: percussion peaks, P2: tidal peaks, and P3: diastolic peaks). This means that our e-skin can detect a whole range of body movements, from very subtle pulse pressure waves to movements of muscles and joints and suggests that the e-skin has the potential for application to fields of bio-healthcare and robot control. As shown in Figure 4f, different waveforms were recorded as the sensor responded to repetitive sounds. The subtle movements of the vocal cords, the difference in the intensity of sound, and the difference in the height of sound appear as different waveforms through the sensor, and it is possible to apply the uniquely formed signals to the Internet of Things (IoT) and other industries that involved voice recognition. In addition, Figure 5 shows the sensing ability of our e-skin to subtle oscillations from vocal cord. The e-skin was attached to the Adam’s apple as shown in Figure 5, monitoring subtle oscillations of Larynx and vocal cord due to saliva swallowing (Figure 5a), sniffing (Figure 5b), coughing (Figure 5c), yawning (Figure 5d), and as well as phonation of four different greetings (Figure 5e–h). The electrical signal records in response to saliva swallowing present two distinct curves for each time. Likewise, the expansion of the bronchi caused by the instantaneous breathing in a large amount applied strain to the e-skin, causing relative resistance changes to appear. In the case of coughing, the subtle movement of the neck is also a huge factor of strain applied to the e-skin, so it induced the relative resistance change to ~40. The e-skin presents the different electrical signal changes for the four greetings: Korean (Annyounghaseyo), Japanese (Konnichiwa), Chinese (Nihao), and English. The sensor showed peaks in the pronunciation where movement of the vocal cords such as “YO”, “A”, “WA”, and “OU” occurs.

Therefore, the peak pattern in signal records were distinguishable from each other, confirming the reliability of the e-skin to detect motion from vocal cords. Most diseases that cause dysphagia are senile nervous system diseases such as dementia and Parkinson’s disease, of which stroke is the most common. The process of swallowing food is controlled by the central part of the brainstem, through the cranial nerves to the oral cavity, pharynx, and larynx. Therefore, finding out whether the subject’s salivation process is doing well can be of great help in diagnosing the aforementioned diseases.

## 4. Conclusions

In summary, based on LIG, strain sensors with a closed-porous structure were fabricated by irradiating a laser onto a PDMS/PSPI substrate using LDW; this is a controlled, cost-effective, one-step, and continuous process. By changing the weight ratio of PDMS to PSPI and the laser scanning speed, the color of the substrate and the working range, respectively, can be tuned. We demonstrated the advantage of introducing hollow structures into the LIG to further increase the air gap volume and reduce the effective mechanical modulus. The fabricated strain sensor shows a high stretchability (over 90%) and high sensitivity (~35-gauge factor) as well as a good linear relationship between the applied strain and electrical resistance change. Furthermore, the skin color-like strain sensor showed small hysteresis and excellent durability under cyclic stretching–releasing cycles. A tremendous number of potential applications for our sensor, in detecting not only subtle oscillation but also vigorous human motions (such as finger, wrist, elbow, and knee bending) were demonstrated. Such novel strain sensors are expected to be applied as next-generation materials for facile integration into smart wearable devices for human motion detection, personalized healthcare, and electronic skin. Our e-skin showed excellent response to subject’s saliva swallowing, which proves the possibility of using the sensor as a diagnostic device.

## Figures and Tables

**Figure 1 nanomaterials-11-00951-f001:**
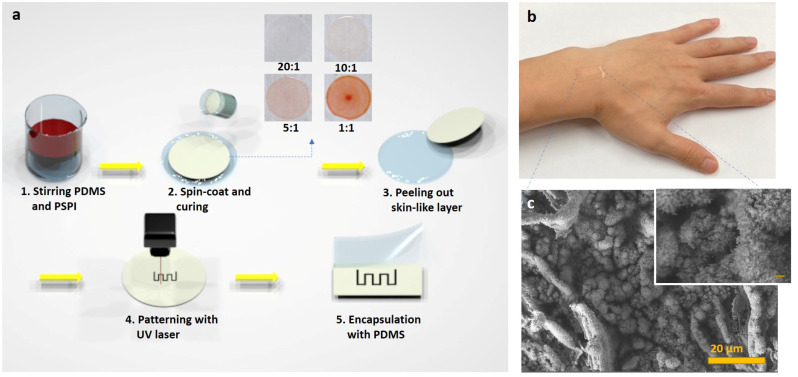
E-skin formed from skin-like polymer using UV pulsed laser irradiation; (**a**) schematic of fabrication process of e-skin from skin-like polymer, (**b**) E-skin placed onto wrist to sense joint movement, (**c**) SEM image of laser induced graphene (LIG) patterned on skin-like polymer. (scale bar: 20 μm).

**Figure 2 nanomaterials-11-00951-f002:**
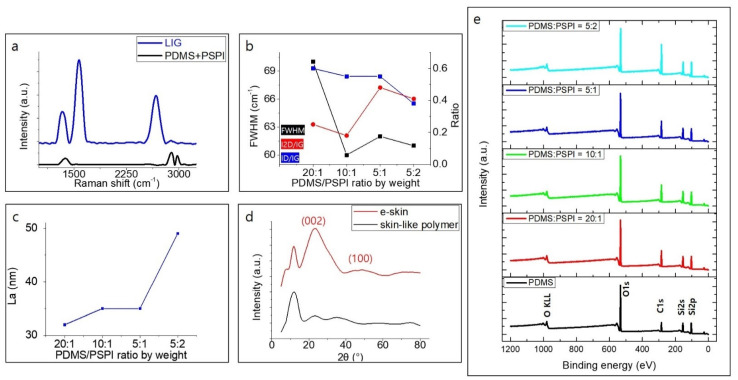
Characterizations of laser-induced graphitic pattern fabricated according to different weight ratio of polydimethylsiloxane (PDMS)/photosensitive polyimide (PSPI); (**a**) Raman spectrum of an LIG pattern and the skin-like polymer, (**b**) statistical analysis of Raman spectra, (**c**) average domain size along the a-axis of graphene aerosol in the LIG pattern, (**d**) the XRD results of e-skin and the skin-like polymer, (**e**) XPS characteristics of the e-skin fabricated by according to the various weight ratio of PDMS/PSPI.

**Figure 3 nanomaterials-11-00951-f003:**
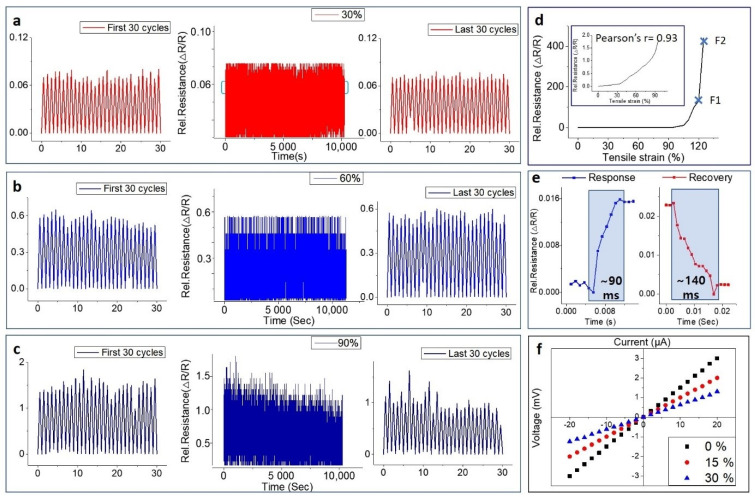
Strain sensing performance of the e-skin; change in relative resistance for (**a**) 30% strain, (**b**) 60% strain, and (**c**) 90% strain. (**d**) linear relative resistance changes from 0 to 120% strain of the e-skin, (**e**) response and recovery time of the e-skin for strain, (**f**) changes in current-voltage (I-V) characteristics of e-skin according to strain.

**Figure 4 nanomaterials-11-00951-f004:**
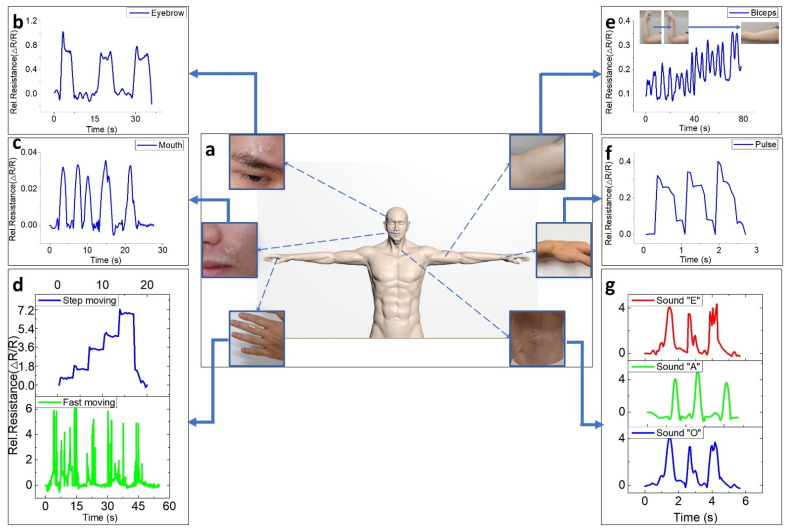
Application of e-skin for wearable devices; (**a**) schematic image of e-skin attached to various parts (eyebrow, mouth, finger, biceps, wrist, vocal cord) of the human body; (**b**,**c**) response of the e-skin for movement of the eyebrow and mouth to sense facial expression; (**d**,**e**) response of e-skin for movement of finger and biceps while stepping and moving fast. (**f**) Sensor response of e-skin for pulse. (**g**) Sensing for the sounds “E”, ”A”, and “O” while attached onto throat (vocal cord).

**Figure 5 nanomaterials-11-00951-f005:**
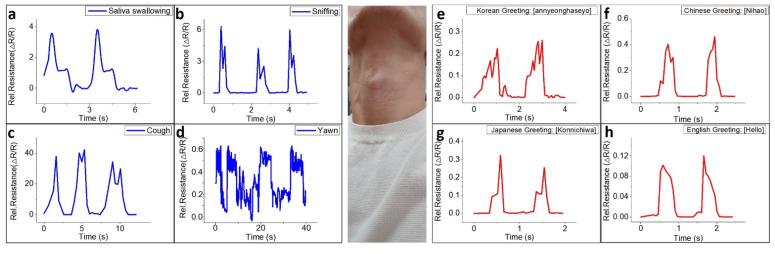
Electrical signal records for subtle oscillations associated with the human vocal cord; the corresponding time dependent electrical signals of (**a**) swallowing salvia, (**b**) sniffing, (**c**) coughing, (**d**) yawning, and phonation of words four greetings: (**e**) Korean, (**f**) Chinese, (**g**) Japanese, and (**h**) English.

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
