# Peer review of "Highly Skin-Conformal Laser-Induced Graphene-Based Human Motion Monitoring Sensor"

_nanomaterials, 2021, doi:10.3390/nano11040951_

Round 1
Reviewer 1 Report
Some minor spelling errors have been found, such as:
- line 65: CO2 (subindex missing)
- line 69: irradiation48
- line 73: hu-man
- line 86: be-cause
- line 164: "355 nm" appears two times
- line 186: "both side" should be "both sides"
I would also recommend to improve the quality of the formulas for an easier reading.
I suggest to comment on the work performed by A. Vashisth et al., “ReaxFF Simulations of Laser-Induced Graphene (LIG) Formation for Multifunctional Plymer Nanocomposites”, ACS Appl. Nano Mater. 3, 1881-1890, 2020, and give their critical opinion on this kind of simulation works.
Could you give a bit more detail on the integration of the proposed fabrication process in a mass production level?
Author Response
Response to Reviewer 1 Comments
Comments and Suggestions for Authors
- Some minor spelling errors have been found, such as:
- line 65: CO2 (subindex missing)
- line 69: irradiation48
- line 73: hu-man
- line 86: be-cause
- line 164: "355 nm" appears two times
- line 186: "both side" should be "both sides"
Response 1: all spelling errors have been revised.
- I would also recommend to improve the quality of the formulas for an easier reading.
Response 2: The way to express formulas has been improved. (Add space between formulas/Improvement of formula expression method) - I suggest to comment on the work performed by A. Vashisth et al., “ReaxFF Simulations of Laser-Induced Graphene (LIG) Formation for Multifunctional Plymer Nanocomposites”, ACS Appl. Nano Mater. 3, 1881-1890, 2020, and give their critical opinion on this kind of simulation works.
Response: (A. Vashisth et al., “ReaxFF Simulations of Laser-Induced Graphene (LIG) Formation for Multifunctional Plymer Nanocomposites”, ACS Appl. Nano Mater. 3, 1881-1890, 2020) has been added to the main text. (Line Nos. 70-73)

Reviewer 2 Report
Authors are kindly invited to check the attached PDF were detailed comments and annotations have been reported.
Additionally, authors are encouraged to check the way they express the results of performed measurements, trying to.
- report the measurement uncertainty for each result, and avoiding using symbols such as ∼ which do not have a clear meaning from e metrological point of view; also the notation used to report measurement results and strain indication should be revised (see attached file for details, thanks);
- about the experiment on voice detection, is the sensor usable also in women who do not have the Adam's apple?
- in line 429, to state "normal heart condition" is not appropriate as no reference instrument has been used to measure the HR and validate the results provided by the e-skin;
- unfortunately, images and video indicated as supplementary material seem to be not accessible.
Author Response
Response to Reviewer 2 Comments
Comments and Suggestions for Authors
Comments and Suggestions for Authors
Authors are kindly invited to check the attached PDF were detailed comments and annotations have been reported.
Additionally, authors are encouraged to check the way they express the results of performed measurements, trying to.
1.report the measurement uncertainty for each result, and avoiding using symbols such as ∼ which do not have a clear meaning from e metrological point of view; also the notation used to report measurement results and strain indication should be revised (see attached file for details, thanks);
Response: We have done round-off some values. For example, round-off: 15.2142 → ~15.
Unfortunately, there is no attached file on the website.
- about the experiment on voice detection, is the sensor usable also in women who do not have the Adam's apple?
Response: Our sensor detects subtle oscillation occurring in the vocal cords, so even if the test subject is a female, the sensor can sufficiently monitor the response.
- in line 429, to state "normal heart condition" is not appropriate as no reference instrument has been used to measure the HR and validate the results provided by the e-skin;
Response: The phrase ",indicating normal heart condition" has been deleted from the text.
- unfortunately, images and video indicated as supplementary material seem to be not accessible.
Response: We have attached new version zip-file.

Reviewer 3 Report
In this submission, Jeong et al. report on a new fabrication technology for building elastomeric conductive skin-like composite by mixing polymer solutions with the formation of laser induced graphene (LIG) on the skin-like substrates to efficiently monitor not only energetic human motions but also subtle oscillation and physiological signals for intelligent sound sensing. This is an interesting study. I recommend this paper to be accepted with subject to minor revisions. Following are my specific comments;
- Figure 1 c – scale bar should be added to the figure.
- Number of layers of graphene should be added in Raman characterization section and also in discussion section.
- Line 229 – ‘The 2D peak is related to the number of and the stacking order of the graphene …’ this sentence need a reference.
- Any role of surface charge on binding and sensing?
- What is specific surface area of the composite? Did authors measure this? The role of surface area in sensing needs to elaborated a bit more.
- I suggest authors to cite a recently published article on this subject; https://doi.org/10.1002/wnan.1699
Author Response
Response to Reviewer 3 Comments
Comments and Suggestions for Authors
Comments and Suggestions for Authors
In this submission, Jeong et al. report on a new fabrication technology for building elastomeric conductive skin-like composite by mixing polymer solutions with the formation of laser induced graphene (LIG) on the skin-like substrates to efficiently monitor not only energetic human motions but also subtle oscillation and physiological signals for intelligent sound sensing. This is an interesting study. I recommend this paper to be accepted with subject to minor revisions. Following are my specific comments;
- Figure 1 c – scale bar should be added to the figure.
Response: We have added scale bar in the figure. - Number of layers of graphene should be added in Raman characterization section and also in discussion section.
Response: LIG has not only hexagonal carbon rings, but also pentagonal and heptagonal carbon rings, so it has a three-dimensional multilayer structure. Therefore, it is not possible to accurately count the number of layers by having a high 2D peak, as it is not a one to three-layer structure that can be confirmed by a Raman spectrum. - Line 229 – ‘The 2D peak is related to the number of and the stacking order of the graphene …’ this sentence need a reference.
Response : Reference No. 55 was added. - Any role of surface charge on binding and sensing?
Response: As shown in Figure S7, when strain is applied to the sensor, not only the surface but also the structure inside the sensor is changed. This reduces the conductive pathway and induces a rapid increase in resistance. - What is specific surface area of the composite? Did authors measure this? The role of surface area in sensing needs to elaborated a bit more.
Response: Unfortunately, we should resubmit revised version within 5 days. So, we don’t have enough time to check result of BET or other surface area related analysis. - I suggest authors to cite a recently published article on this subject; https://doi.org/10.1002/wnan.1699
Response: Reference No.13 was added.
